# A Weighted and Normalized Gould–Fernandez brokerage measure

**Zsófia Zádor**[1]*, **Zhen Zhu**[2], **Matthew Smith**[3], **Sara Gorgoni**[1]

**1** Department of International Business and Economics, University of Greenwich, London, United Kingdom, **2** Kent Business School, University of Kent, Canterbury, United Kingdom, **3** The Business School, Edinburgh Napier University, Edinburgh, United Kingdom

* zador.zsofi@gmail.com

**Data Availability Statement:** All EUREGIO files are available from the EUREGIO database, that can be publicly accessed at https://data.overheid.nl/en/dataset/pbl-euregio-database-2000-2010 All Organizational network files are available from Tore

## Abstract

The Gould and Fernandez local brokerage measure defines brokering roles based on the group membership of the nodes from the incoming and outgoing edges. This paper extends on this brokerage measure to account for weighted edges and introduces the Weighted–Normalized Gould–Fernandez measure (WNGF). The value added of this new measure is demonstrated empirically with both a macro level trade network and a micro level organization network. The measure is first applied to the EUREGIO inter-regional trade dataset and then to an organizational network in a research and development (R&D) group. The results gained from the WNGF measure are compared to those from two dichotomized networks: a threshold and a multiscale backbone network. The results show that the WNGF generates valid results, consistent with those of the dichotomized network. In addition, it provides the following advantages: (i) it ensures information retention; (ii) since no alterations and decisions have to be made on how to dichotomize the network, the WNGF frees the user from the burden of making assumptions; (iii) it provides a nuanced understanding of each node's brokerage role. These advantages are of special importance when the role of less connected nodes is considered. The two empirical networks used here are for illustrative purposes. Possible applications of WNGF span beyond regional and organizational studies, and into all those contexts where retaining weights is important, for example by accounting for persisting or repeating edges compared to one-time interactions. WNGF can also be used to further analyze networks that measure how often people meet, talk, text, like, or retweet. WNGF makes a relevant methodological contribution as it offers a way to analyze brokerage in weighted, directed, and even complete graphs without information loss that can be used across disciplines and different type of networks.

## Introduction

Brokerage is commonly defined as the process through which an actor connects otherwise disconnected actors or groups [1] and it is a key concept in Social Network Analysis (SNA). The origin of the concept can be traced back to the discussion of third-party influence by Simmel [2], and Granovetter's concept of the strength of weak ties [3] where weakly connected nodes may be able to transfer information that is inaccessible through strong ties. This powerful

Opsahl's dataset Network 8-11, that can be publicly accessed at https://toreopsahl.com/datasets/#Cross_Parker.

**Funding:** The author(s) received no specific funding for this work.

**Competing interests:** The authors have declared that no competing interests exist.

concept has applicability to a variety of domains, and for this reason it has been investigated in the context of various disciplines, including sociology, management, and economics [4–6]. Despite the wealth of applications to date there are only a handful of measures that consider edge weights. In this paper, we aim to contribute to the brokerage literature by extending the Gould–Fernandez [4] (GF) brokerage measure to account for edge weights and normalization. The main contribution of this extension is to allow the GF analysis of weighted networks without changes or manipulation of the observed data, preserving the network structure and the existing heterogeneity in the topology of the network.

This paper demonstrates through two empirical applications that the novel Weighted and Normalized Gould–Fernandez measure (WNGF) detects salient features of networks that would otherwise remain concealed. It does so by taking into account all available information for directed and weighted graphs. The need for weighted versions of established network measures has also been identified by other scholars, with the extension of betweenness centrality [7] and, more recently, with the extension of page rank centrality [8].

Normalization is also implemented in the WNGF to account for any bias arising from differences in group size. While there are instances where a non-normalized metric could be better, for example because a broker (for instance a gatekeeper) in a large group could be more relevant than a broker in a small group, in other cases normalization may improve comparison across groups of different size when we want to account for the size. An example of this is when we wish to identify the most important brokering role for a node in comparison to all others. In order to demonstrate the feasibility and value of using the WNGF, we apply it to two different networks. First, we apply it to the EUREGIO regional input–output tables, from which a complete, weighted, and directed graph can be constructed. Second, we compute it on an organizational network of an R&D group that has a smaller, less dense structure. The results of the application of the WNGF measures are compared to two dichotomization procedures. Network dichotomization is most often accomplished through the threshold approach. Hence, we choose threshold dichotomization at 10% and 35%, respectively, for the two datasets. Second, the multiscale backbone method [9] is selected as a common tool for dichotomizing economic networks [10, 11]. Various cut-off (alpha) values are explored, which ensure that all the nodes are kept in the network and the arbitrary value of 0.4 cut-off point is chosen. More details on this are reported in Section 3. The results show that while the WNGF measure produces similar results to those of the dichotomized ones, it still retains more information, and allows for a more nuanced understanding of the brokerage roles played by each node in the network.

The paper is structured as follows: in the next section the literature on brokerage is discussed, with special focus on the Gould–Fernandez measure. It is then followed by an overview of the various applications of brokerage measures in the empirical literature. In the Methodology Section the Weighted–Normalized GF (WNGF) measure is introduced, followed by the current dichotomization techniques. Next, we compare the non-weighted Gould–Fernandez and the WNGF measures by applying them to the weighted and directed graph transformed from the EUREGIO trade data and the organizational network. The last section discusses the results and provides a conclusion.

## Theoretical background

### Brokerage

In his landmark book *Structural Holes* [12] Ron Burt argues that ties that bridge holes in social structure provide important advantages, such as access to new information. In his later book *Brokerage and Closure* [13] he further develops this argument, stating that brokers will have

three types of advantages: (i) access to more diverse information, which allows them to be more creative; (ii) earlier access to information; and (iii) control over information flows. The two concepts of brokerage and structural holes are tightly interconnected, as a broker who connects unconnected others essentially fills in holes in the network [4, 12].

The sociological interest in brokerage [2, 3] centers around the core argument that brokerage facilitates control over resources between unconnected actors and that the broker is likely to benefit from this, for example by becoming more influential and powerful [14], or by experiencing enhanced economic activity [15–17]. Stovel and Shaw [18] also discuss how brokerage is the result of an uneven distribution of resources, and how it can in turn exacerbate inequalities; how it can maintain group boundaries, but also integrate communities.

There are various ways of measuring brokerage; the most common node-based measure that takes into account the whole network structure is betweenness centrality. Betweenness centrality is the most established method for understanding brokerage, and it has been modified and extended in numerous ways. Betweenness centrality was first defined by Freeman [19] based on a study of human communication, while Bavelas [20] discovered that the position of some people proved to be strategic as they connected others in the group directly. Since then, understanding various aspects of node positions became an essential part of SNA. Betweenness centrality ranks nodes based on the number of times node $v$ (where $v \in V$) lies on the shortest path ($\sigma_{st}$) from node $s$ to node $t$. The node with the highest betweenness centrality is the one that most often lies on the shortest path connecting others, thus can reach other nodes the fastest.

Various measures of betweenness centrality have been developed over the past decades, modifying the original measure to better capture various dimensions. Betweenness centrality was extended for weighted graphs [7] and used to create an edge betweenness divided by degree [21]. Several betweenness measures have also been created based on different characteristics of networks, such as network flow [22], or random walks [23]. Betweenness centrality, as a global network measure which accounts for indirect connections as well as direct ones, can shed light on important aspects of the overall network; for example, how vulnerable the network is to attack can be dependent on the number of brokers in the network [21]. The practicality of betweenness centrality stems from its ability to account for edge weights, with this providing more detailed and complete understanding of the topology of complex graphs [24]. However, a shortcoming of traditional betweenness measures is that they take a static network into account, which may not be ideal when dealing with information or disease and so on spreading in a network. Therefore, the percolation literature has been particularly interested in establishing centrality measures that take the changing percolation state into account [25]. Sciabolazza and Riccetti [26] define the novel diffusion delay centrality (DDC) measure that relies only on the network structure, betweenness, and eigenvector centrality to find the most central nodes in terms of diffusion.

Alongside global brokerage measures like betweenness centrality, which locate the brokers with respect to all the other nodes in the network, local brokerage measures are also widely established in the literature. Examples of local brokerage measures are Burt's [12] constraint, based on the triadic closure principle, and Gould and Fernandez (GF) brokerage roles. The GF brokerage examines a node's relations with its immediate neighbors from the perspective of the node, by finding every instance where that node lies on the direct path between two others [27, 28]. Gould and Fernandez [4] show that there are five structurally distinct types of brokers that follow from a partitioning of actors into non-overlapping subgroups. GF brokerage can be considered a constrained version of betweenness centrality as it only counts two-step paths between nodes assigned to different groups. By classifying nodes into groups, the Gould-Fernandez brokerage roles place special importance on group boundaries and dynamics. The five

GF roles can be differentiated from each other based on the node attributes that help categorize them into groups, even if they share the same structure. A broker can be: (i) a *liaison*, if they mediate between members of different groups; (ii) a *representative*, when the broker connects their own group to another group; (iii) a *gatekeeper*, when the broker acts as a gatekeeper for their group and can decide whether or not to grant access to an outsider; (iv) an *itinerant* (or cosmopolitan), if the broker mediates between members of a group they do not belong to; and, finally, (v) a *coordinator*, enhancing interaction between the members of the group the broker belongs to. Therefore, the GF brokerage measure goes beyond a single metric to examine what type of brokering function a focal node plays within a network. Consequently, the GF measure takes an ego perspective to brokerage, where it uses directed data by definition. Therefore, it is useful in situations where directionality and membership are important features, and where direct neighbors are thought to contribute the most to the benefits of becoming a broker. This measure allows identification of the source and mapping of the route of resource flows, which in turn allows for a more nuanced understanding. The ability to define the group membership further enhances the understanding of who the broker connects. The GF brokerage measure has been adapted and extended in recent years [29]; for instance, Jasny and Lubell [30] developed a version of the GF brokerage roles that could be applied for two-mode networks. However, the GF brokerage measure has not been extended for the case of weighted networks.

## Literature review of the applications of the GF brokerage measure

Brokerage is traditionally a micro level concept that finds applications in a variety of social science disciplines [28]. In various social settings individuals act as middlemen in order to, for example, bring people with diverging political views together in politics [4, 31, 32], coordinate transactions in economics [33, 34], or facilitate the relocation or integration of new arrivals in migration studies [18, 35–37]. Besides sociology, one of the most fertile areas of research on brokerage has been management and organization studies. On the micro level, this literature explores how an individual might advance in their career [38], or is more prone to creative ideas [5] due to their structural position. On the meso level the advantages—such as innovation performance—to brokering firms compared to non-brokering firms [39], as well as the potential disadvantages, [40] are emphasized. Macro level brokerage is more frequently covered in regional and international trade studies, where different spatial levels, cities, regions, or countries are considered for their brokering position.

Applications of GF brokerage equally span across different disciplines. For example, Kirkels and Duysters [41] aim to understand the variation in brokering knowledge flows in small to medium-sized enterprise (SME) networks and thus turn to a GF brokerage analysis. In their ground-breaking research on how networks influence organizations, Cross and Parker [42] conducted surveys on advice flow, trust in advice, and knowledge of skills on a Likert scale, but in the analysis they had to disregard the information on edge weights due to the lack of a method that could handle valued data. With the increasingly abundant availability of valued data, it becomes imperative to develop proper analytical tools to deal with the rich information that they bring.

More recently, the concept of brokerage has also been investigated in the area of economic geography, exploring the effect of knowledge exchange between countries, regions, and cities. In these studies, focus is generally on the access to global, relevant, or differentiated resources from that available in the region or innovation cluster [43, 44]. While brokerage is becoming a more prevalent concept in this literature, only a few papers investigate the varying effect of different brokerage types. Drawing on the need to understand the heterogeneity of brokers, Seo [45] examines what kind of technology broker is needed for economic development; Sigler

et al. [46] look at city-regions in the context of global corporate networks, and by using the GF brokerage measure find that city-regions confer economic advantage through their brokerage roles. Martinus et al. [47] investigate how small states and territories act as brokers in the global corporate network. GF brokerage has also been increasingly applied in international trade studies in the context of the international fragmentation and complex organization of production [6, 48]. Gorgoni et al. [6] in their examination of the automotive sector apply the GF brokerage roles to analyze the different ways in which countries act as brokers between different global regions (e.g. Europe, North America). Along the same lines, Smith and Sarabi [48] analyze the role of the UK in brokering within the EU region and between the EU and other global regions. In both studies, a weighted and directed network is constructed first. However, since there is no current method to analyze brokerage topology on weighted networks, the networks are dichotomized. This highlights the need for extending current measures to handle valued network data. To the best of our knowledge, none of the proprietary (Ucinet and Pajek) or open-source software for social network analysis has an algorithm that allows computation of the GF brokerage measure to account for weighted data. For this reason, we believe that the WNGF provides an important and much awaited methodological contribution. In this paper we have implemented the WNGF in Python.

## Methodology

### Introduction of the weighted–normalized Gould–Fernandez measure

The modified algorithm for the WNGF identifies the different brokerage roles that a node in a weighted network might assume. The theoretical underpinnings of the weighted GF measure are rooted in the weighted shortest path literature [49, 50]. For unweighted networks, the calculation of shortest path is straightforward. In an unweighted network, the shortest path from node $q$ to $s$ is the path with the least number of intermediate nodes. Shortest paths calculations for weighted networks are most common in ecological and transport networks, where the weight identifies the length, distance, or cost of the road, dispersal time or frequencies, shared traits, and flows in interaction networks. In such networks the length of the path is calculated by summing up the edge weights along the path. The aim is to find the path with the least resistance or the smallest weight to channel the information. As Costa et al. [51] describe in detail for ecological networks, for some networks finding the least resistance means that edge weights have to be transformed. For those edge weights, where the weights are proportional to the strength of the relationship between nodes, edges need to be reversed.

In parallel with the weighted shortest path literature, for each node, we consider its incoming and outgoing neighbors in the network. In this context, we define brokerage as follows: if the sum of the inverse of the transaction flow between the focal node and either neighbor is smaller than the inverse of the flow directly from the incoming to the outgoing neighbor, then the focal node is identified as a broker. For example, node $r$ will be identified as a broker from node $q$ to node $s$ if the relationship below is true.

$$\frac{1}{Z_{qr}} + \frac{1}{Z_{rs}} < Z_{qs},\tag{1}$$

where $Z$ is the flow between the broker node $r$, and it's neighboring nodes $q$ and $s$. Note that $Z_{qs} = 0$ if there is no edge between node $q$ and node $s$.

The type of brokerage role is then determined by the group memberships of all three nodes, as described below. Fig 1 visualizes the different roles and provides the notation for each of the five roles and their normalized version. For example, $Z_{rs}^{ij}$ is the flow from node $r$ in group $i$ to node $s$ in group $j$.

| Brokerage Role | Visualization | Description | Normalizing Denominator |
|---|---|---|---|
| Coordinator | | $\dfrac{1}{Z_{qr}^{ii}} + \dfrac{1}{Z_{rs}^{ii}} < \dfrac{1}{Z_{qs}^{ii}}$ | $n_b^i = 2\dbinom{m^i - 1}{2}$ |
| Gatekeeper | | $\dfrac{1}{Z_{qr}^{ji}} + \dfrac{1}{Z_{rs}^{ij}} < \dfrac{1}{Z_{qs}^{ji}}$ | $n_b^i = \sum_{j,j\neq i} m^j (m^i - 1)$ |
| Representative | | $\dfrac{1}{Z_{qr}^{ii}} + \dfrac{1}{Z_{rs}^{ij}} < \dfrac{1}{Z_{qs}^{ij}}$ | $n_b^i = \sum_{j,j\neq i} m^j (m^i - 1)$ |
| Itinerant | | $\dfrac{1}{Z_{qr}^{ji}} + \dfrac{1}{Z_{rs}^{ij}} < \dfrac{1}{Z_{qs}^{ii}}$ | $n_b^i = \sum_{j,j\neq i} 2\dbinom{m^j}{2}$ |
| Liaison | | $\dfrac{1}{Z_{qr}^{ji}} + \dfrac{1}{Z_{rs}^{ik}} < \dfrac{1}{Z_{qs}^{jk}}$ | $n_b^i = \sum_{j}\sum_{k,k\neq j, j\neq i, k\neq i} m^j m^k$ |

**Fig 1. Visualization and description of GF brokerage roles.** *Note: The square is the broker, node color indicates group. Authors' elaboration based on Gould and Fernandez* [4].

In the last column of Fig 1, the normalization of each broker role is defined. While normalization may not always be necessary, it is crucial if the aim is to compare the importance of brokerage roles across nodes, as well as across different roles of the same node. The size of groups may vary, as is the case in both of our networks. Not accounting for the difference in the number of group members would introduce a bias in some cases towards groups with a higher member count, while in other cases towards groups with a lower member count. To illustrate, a hypothetical example will be presented. We want to identify the representative and coordinator roles from a dataset with two groups: Group A with three members and Group B with ten. The maximum times a Group A node—let's call it A1—can be a representative is 20 (connecting all other nodes in Group A to all ten Group B nodes), while only twice can it be a coordinator. While this difference matters in terms of transaction costs, does it mean that the coordinator role of A1 is less important that the representative role? On the other hand, a node —let's call it B1—in Group B can be a representative 27 times (connecting all nine Group B nodes to all three Group A nodes), and a coordinator 72 times (all nine nodes in Group B can

be connected to all remaining eight nodes). Does the fact that B1 can be a coordinator 72 times mean that it is less fully connecting Group B than A1's two coordinator role? This example highlights the potential bias without normalization and how it would affect the various GF roles differently. Since the aim here is not to address the transaction cost of brokerage but to understand how important each node's brokering position is within the network as a whole, GF role-specific normalization is introduced. This way, not only can we safely compare nodes and type of roles to each other, we can access the importance of a node to carry out the different types of broker positions. While in the context of this paper normalization is relevant for the reasons discussed, it is worth noting that it is a function implemented in many existing GF packages, including in Ucinet and Pajek. For a brokerage type $b$, let $c_r^i$ be the number of times node $r$ in group $i$ assumes the role, and $n_b^i$ be the number of possible brokered relationships, then its normalized brokerage importance is computed as:

$$p_r^i = \begin{cases} \dfrac{c_r^i}{n_b^i}, & n_b^i > 0 \\ \\ 0, & n_b^i = 0 \end{cases} \tag{2}$$

Therefore, $p_r^i$ gives us the proportion of node $r$ in group $i$ to assume a role, computed by dividing the counted brokerage roles over all possible roles. This calculation can only take place if the possible brokering roles is not 0. Note that $n_b^i$ varies across groups as well as types of brokerage. For each type of brokerage, $n_b^i$ represents the theoretical upper bound of the number of times a node can be identified as a broker. Therefore, the normalized brokerage importance $p_r^i$ is bounded between 0 and 1. Let $m^j$ be the number of nodes in group $j$.

## Current dichotomization techniques

Dichotomizing weighted networks is a widely used approach, with increasingly accurate methods that ensure that the structure of the network is kept mostly intact. The main weaknesses of these practices, however, is that the network topology is still altered at least to some extent. In addition, removing edge weights, and edges based on their weights or importance, inherently leads to information loss. Edge weights typically present a challenge to SNA scholars, as several models and metrics, including the GF brokerage measure, are only available for binary or unweighted networks [52]. This often results in the implementation of a dichotomization of the network, transforming the weighted network into an unweighted one. Extant work has considered this issue in detail [53–55], and there are a wide number of algorithms and techniques available [56]. The simplest is to disregard all the edge weights, yet retain all the edges; however, this does not account for the heterogeneity of edge weights. Additionally, this approach is only useful when the density of the network is relatively low; if the weighted network is dense with a high level of connectivity, this approach proves less effective [52].

A further popular approach is to apply a threshold, that is to take a particular value (often an arbitrary value) and disregard all edges with weights less than this value [57]. A challenge with this approach is selecting the appropriate threshold value, as a network with an uneven distribution of edge weights (which is frequently observed in real world networks) can amplify the issues of arbitrariness and structural bias of the threshold approach [58]. Attempts have been made to develop more robust approaches in the selection of the threshold value, for instance both Derudder et al. [59] and Dai et al. [55] suggest taking a threshold value that results in a network with a density level that most closely resembles the original network, while Giordano and Primerano [60] also rely on further statistical characteristics, specifically those used to determine information on the components of the network to reliably determine the

most appropriate threshold level. Besides density, they also investigate the number of components and betweenness centrality.

Zhou et al. [61] propose an alternative threshold approach. They suggest that instead of taking a network-wide threshold, the top edges for each actor should be retained. For instance, the top three edges (by edge weight) for every actor in the network would be retained. Zhou et al. [61] carry out an empirical analysis on the international trade network, retaining the top trade relations for each country. They argue that this approach guarantees the inclusion of all nations in the network and allows to control for the density of the network. However, a potential issue with this approach is that it treats the top tie of a periphery country in the trade network the same as the ties of active trading nations (such as the USA or China).

In addition to the threshold dichotomization procedures, there are various techniques for assessing an edge's significance, and to use this as the basis for the edge reduction. Serrano et al. [9] propose a more systematic approach to extract the key edges from a network, which they refer to as the network backbone and is most commonly referred to as the disparity filter. This is a filtering approach which removes all non-essential links, retaining the significantly heterogeneous links at specified significance level. This approach helps retain key ties, whilst preserving the overall connectivity of the network. The multiscale backbone filtering procedure has become an established approach within network science and, while the main aim of the tool was not to dichotomize the network, there are numerous applications in international trade [10, 11], transport [62], and a variety of other empirical settings that take advantage of this function of the disparity filter. Backbone extraction methods are also popular with bipartite network projection but may lead to substantial differences depending on the chosen significance level [58], hence a further method was developed to identify the significant links to be retained based on the heterogeneity of all three nodes involved in a tri-partite migration network without having to rely on an arbitrary significance level [63].

In order to analyze the effectiveness of using the GF method on the dichotomized network we will compare two different dichotomization techniques, one based on a threshold level [57] and one based on creating a network backbone [9]. For the former, we consider a threshold where 10% of the bottom edges are removed for the EUREGIO network, while for the organizational network 35% of the edges are cut. Different levels are chosen as the second network has categorical values as weights, and all the edges with a weight of 1, the least relevant category, are removed. To create a multi-scale network backbone, we take the significance level $\alpha$ = 0.4, under which we preserve the edges. These levels were chosen arbitrarily after exploring the distribution of various values. We explore the different levels as long as all nodes were retained in the network. The node retention criterion was the main consideration when choosing the cut-off points. All nodes of the network need to be retained in order to compare their brokering role in a dichotomized GF against the WNGF method. Second, we explore the heterogeneity of the various dichotomized networks and aim to choose a cut-off point that reserves most of the heterogeneity in terms of the brokering role. To evaluate closeness to uniform distribution, Empirical Cumulative Distribution Functions (ECDFs) are created. Lastly, we rely on the standard practice in the literature for selecting the cut-off point.

For threshold dichotomization, we explored six different levels between 5%–60% and found that until the 40% threshold, the higher the threshold the more uniform the distribution is, while lower threshold levels underrepresented low brokering values. As removing 40% of the edges is far from standard practice, the 10% threshold is chosen. For the multiscale backbone, we explored $\alpha$ measures up to 0.5 that were claimed optimal [9]; however, the lowest $\alpha$ measure that still retains all nodes is $\alpha \geq 0.34$. As these distributions are relatively close to each other, we chose the standard $\alpha$ = 0.4 significance level. Interestingly, a similar trend can be observed as with the threshold levels, where the larger the $\alpha$ the more uniform the distribution;

however, here lower $\alpha$ levels overrepresented low brokering values. We present the decision-making process here to highlight that as soon as dichotomization takes place, arbitrary decisions have to be made without solid methodological guidelines to rely on regarding the appropriate levels. Furthermore, as highlighted by Borgatti and Quintane [57], dichotomization by definition removes information and distorts data. Our aim in this paper is not to find the best way to dichotomize the network, instead to present a reliable, granular method that retains all the information, without having to make arbitrary decisions.

## Data description and results

### Overview of the data

In this section, the GF brokerage metrics are computed on two very different type of networks: the first is based on the EUREGIO dataset [64], which is an almost complete network where edge weights play a crucial role in understanding the connections; the second is a network of the advice flow in a distributed R&D group from Cross and Parker [42], which is smaller and less dense but for which in-depth information was available. We show that for both these empirical networks the WNGF measure produces similar results to that of the dichotomized networks but with more in-depth understanding of the role all nodes play.

### Trade network description

The EUREGIO database is the first set of global input–output tables at NUTS2 statistical level for the entire large trading bloc of the European Union covering the years 2000–2010. For each year available, the input–output (trade) flow from each NUTS2 region (or country outside of the EU) to another for each of the 14 sectors is recorded. In total there are 266 NUTS2 regions and countries and therefore 3724 (266×14) units of observations at sectoral level available in each year's input–output table. The numbers in the EUREGIO database are in current basic (producers') prices and are expressed in millions of US dollars.

Table 1 below shows a simplified input–output table with only two countries and four regions, which albeit demonstrates the structure of the EUREGIO database at regional level. The $4 \times 4$ inter-regional component (highlighted in the dashed box) is called the transactions matrix and is often denoted by matrix $Z$. Note that the diagonal domestic submatrices (transactions between regions in the same country) are shaded while the off-diagonal submatrices are the transactions crossing country borders. The rows of $Z$ record the distributions of the industry outputs throughout the two countries while the columns of $Z$ record the composition of inputs required by each region. For example, $Z_{rs}^{ij}$ is the transaction flow from region $r$ in country $i$ to region $s$ in country $j$. Note that in this example all the regions buy inputs from themselves (e.g. $Z_{qq}^{ii}$), which is often observed with significant amounts in aggregated data.

**Table 1. A simplified inter-regional input–output table.**

| | | Country $i$ | | Country $j$ | | Final Demand | Total Output |
|---|---|---|---|---|---|---|---|
| | | **Region $q$** | **Region $r$** | **Region $s$** | **Region $t$** | | |
| Country $i$ | Region $q$ | $Z_{qq}^{ii}$ | $Z_{qr}^{ii}$ | $Z_{qs}^{ij}$ | $Z_{qt}^{ij}$ | $f_q^i$ | $x_q^i$ |
| | Region $r$ | $Z_{rq}^{ii}$ | $Z_{rr}^{ii}$ | $Z_{rs}^{ij}$ | $Z_{rt}^{ij}$ | $f_r^i$ | $x_r^i$ |
| Country $j$ | Region $s$ | $Z_{sq}^{ji}$ | $Z_{sr}^{ji}$ | $Z_{ss}^{jj}$ | $Z_{st}^{jj}$ | $f_s^j$ | $x_s^j$ |
| | Region $t$ | $Z_{tq}^{ji}$ | $Z_{tr}^{ji}$ | $Z_{ts}^{jj}$ | $Z_{tt}^{jj}$ | $f_t^j$ | $x_t^j$ |
| Value-Added | | $v_q^i$ | $v_r^i$ | $v_s^j$ | $v_t^j$ | | |
| Total Output | | $x_q^i$ | $x_r^i$ | $x_s^j$ | $x_t^j$ | | |

Besides intermediate use, the remaining outputs are absorbed by the additional columns of final demand, which includes household consumption, government expenditure, and so forth (here we only show the aggregated final demand for each region). For example, $f_q^i$ is the final demand provided by region $q$ in country $i$. Similarly, production necessitates not only inter-sector transactions but also labor, management, depreciation of capital, and taxes, which are summarized as the additional row of value-added. For example, $v_s^j$ is the value-added required by region $s$ in country $j$. Finally, for each region, the total output can be computed by summing up either its corresponding row or column, which is denoted by $x$.

As in Cerina *et al.* [65] and Riccaboni and Zhu [66], a simple way to construct networks from the input–output tables in the EUREGIO database is to consider the transactions matrix **Z** as an adjacency matrix (e.g. the dashed box in Table 1). Note that the original database has **Z** at sectoral level; however, we concentrate on the trade carried out in the various manufacturing sectors. To do this, we take the five sectors within the manufacturing industry and discard the rest of the sectors. The five sectors are: Food, beverages and tobacco; Textiles and leather etc.; Coke, refined petroleum, nuclear fuel and chemicals etc.; Electrical and optical equipment and Transport equipment; and Other manufacturing. First, we aggregate the manufacturing sector transactions matrix **Z** to regional level before using it as an adjacency matrix. Therefore, taking matrix **Z** as an adjacency matrix creates a weighted, directed network on the manufacturing trade of European regions. For example, the transaction flow $Z_{rs}^{ij}$ of matrix **Z** will become a directed edge from node $r$ (in country $i$) and node $s$ (in country $j$), representing that with the given amount region $r$ is exporting manufacturing products to region $s$.

The inter-regional manufacturing trade network generated from the **Z** matrix is close to a complete, weighted and directed graph, with 266 nodes and 70476 edges, after removing the self-loops. Hence, with an average in and out degree of 264.95, the EUREGIO network falls 14 edges short from being a complete graph. The nodes are NUTS2 regions in the EU, except for 16 nodes which are non-EU and are thus represented as countries. Two nodes are connected if there is manufacturing trade occurring between the two entities.

**Results from the trade network.**   Here we explore the relationship between the results produced by the WNGF and GF methods on the dichotomized network. We do this to understand whether the results obtained are similar to each other, and where the main differences arise from. First, we explore the correlation between the various measures through Pearson and Spearman correlation, then identify the regions with the largest differences between the WNGF and dichotomized measures.

We compare the extent to which WNGF results are correlated with the results obtained from the 10% threshold and the $\alpha = 0.4$ backbone network. Table 2 summarizes both the Pearson and Spearman correlations between the various measures. Table 2 shows that in general, there is a strong correlation between the WNGF results. This suggests that taking into account the full available information and calculating WNGF may be a viable tool to produce valid results on brokerage. Both Pearson and Spearman values show an especially high correlation

**Table 2. Correlation analysis of WNGF results with the backbone and threshold network on the regional trade network.**

|           |          | Coordinator | Liaison | Itinerant | Gatekeeper | Representative |
|-----------|----------|-------------|---------|-----------|------------|----------------|
| **Backbone** | **Pearson** | 0.743*** | 0.855*** | 0.808*** | 0.717*** | 0.638*** |
|           | **Spearman** | 0.704 *** | 0.897*** | 0.907*** | 0.747*** | 0.714*** |
| **Threshold** | **Pearson** | 0.927*** | 0.999*** | 0.993*** | 0.988*** | 0.950*** |
|           | **Spearman** | 0.910*** | 0.999*** | 0.998*** | 0.987*** | 0.944*** |

*** p< = .001

**Table 3. Ranking of the top five regions in terms of largest difference between WNGF and backbone.**

|  | Coordinator | Liaison | Itinerant | Gatekeeper | Representative |
|---|---|---|---|---|---|
| **0.4 backbone** | Stuttgart (DE) | Upp-Bavaria (DE) | Upp-Bavaria (DE) | Attica (EL) | Attica (EL) |
|  | S-Denmark | Stuttgart (DE) | Stuttgart (DE) | South-Finland | South-Finland |
|  | Upp-Bavaria (DE) | Darmstadt (DE) | Paris (FR) | Ctr Macedonia (EL) | Mazovia (PL) |
|  | Attica (EL) | Paris (FR) | Darmstadt (DE) | Mazovia (PL) | Central Hungary |
|  | Darmstadt (DE) | Karlsruhe (DE) | Dusseldorf (DE) | Lisbon (PT) | Stockholm (SE) |

between the WNGF and the 10% threshold network. The similarity between the two results is further highlighted in S1 Fig which, through an ECDF, presents the extent to which each method provides heterogeneous results. The more uniform the distribution, the more heterogeneity there is in the extent to which nodes become brokers. For the liaison, gatekeeper, and representative roles we can see a strong overlap in how the values of the threshold and WNGF network are distributed. Furthermore, in these three roles the distribution is close to uniform. The coordinator and itinerant roles also show that the two network distributions are similar; however, the threshold network tends more toward uniform distribution.

While there is a strong correlation between the WNGF and the dichotomized measures, there are also some differences. Table 3 shows the regions with the largest differences between the two methods. A closer examination of the top five provides interesting insights. For example, Stuttgart, South-Denmark, Upper-Bavaria, and Darmstadt are among the top regions in the role of coordinator according to the backbone network, but not according to the WNGF. If we look at the liaison role, the ranking of the regions in the two measures is very similar; nevertheless, the differences are large.

We see more noteworthy differences in the representative role, where Attica has the largest difference, with the WNGF model returning extra 2991 representative roles. The backbone GF only returns 11 bridging roles for Athens between four Greek regions and Cyprus, Bulgaria, and Turkey. The backbone measure also returns a small number of representative roles for Southern Finland, with Helsinki's region bridging mostly with Russia, China, Estonia, and Lithuania, but also several regions in Sweden. Mazovia in the backbone network seems to be bridging with regional nodes, such as regions in Hungary, Czechia, Slovakia, Austria, and Latvia. These examples highlight the relevance of proximity in retaining edges. Incidentally, proximity is also used as an instrument for trade [67]. Therefore, it seems intuitive that trade within the country and with regional neighbors is retained in the backbone network. While the backbone seems to accurately identify the regional connections, the WNGF measure also identifies the brokerage role of these regions on a more nuanced level.

The regional proximity prioritizing aspect is also shown in terms of the backbone network's gatekeeper role. Here we find Athens bridging from the same three countries (Cyprus, Bulgaria, Turkey), Southern-Finland bridging from Russia, China, and Estonia, but also Copenhagen (DK), The Hague (NL), and several regions in Sweden, while Central Macedonia becomes a gatekeeper only four times in the backbone network from Turkey. These results show that current dichotomization techniques and the WNGF are valid tools to assess the GF roles of regions, and they provide consistent results. However, they convey different benefits and, depending on the purpose of the analysis, one may want to use one or the other. If the aim of the analysis is to keep the most statistically significant edges to understand the underlying processes of a dense network, without too much noise, the backbone measure is the best tool. However, if one is interested in the role of each single node, including those which are more loosely connected, the WNGF measure may be a better solution. For example, an application

of the WNGF on the EUREGIO data would allow the role of each region in countries that do not carry out manufacturing trade in large amounts to be studied. The V4 nations may want to assess the role their regions play in brokering, but would not gain full insight if many of the edges were removed.

## Organizational network description

The second network we apply the WNGF measure to is an organizational dataset collected by Cross and Parker from the researchers of a manufacturing company (data available from [68]). The network contains information on the frequency of advice received from other members of the organization. The network consists of 77 nodes and 2228 edges. There is also some information attributed to the research based on where they are located (Paris, Frankfurt, Warsaw, or Geneva), and their level in the organization (blue: Global department manager; orange: Local department manager; green: Project leader; and pink: Researcher). The edges also have weights from 1 (Very infrequent advice) to 6 (Very frequent advice). Thus, two nodes are connected with a weight of 6 if one receives very frequent advice from the other, where the in-degree is of the advice receiving employee. Fig 2 highlights that employees mostly seek advice from their co-workers based in the same location, therefore four clusters are formed around the four locations of the company.

**Results from the organization network.** From this network we compute the WNGF roles, where we take location as group membership. Therefore, we have four groups for the four countries, across which employees may be providing advice to each other. Understanding who brings in or gives out advice from within a group is valuable, for example in identifying bottlenecks. Fig 2 visualizes the total network, where node color represents the organization level, node size represents number of degrees, node label is the ID of the person, and node

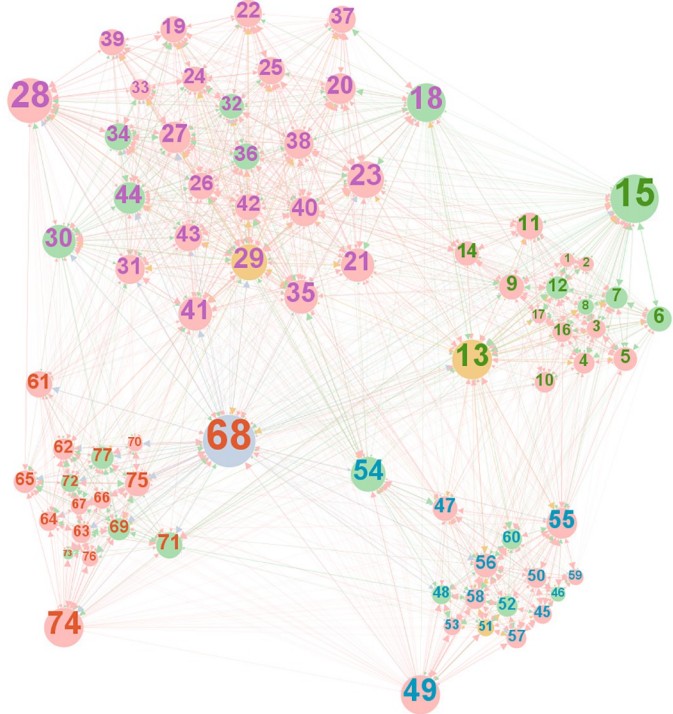

**Fig 2. Organizational network of the R&D section of a manufacturing company.**

label color represents the location. The network shows high clustering among co-workers in the same location, while only a small number of people are in a brokerage position between the various organizations.

First, we use the description of the company in the Cross and Parker [42] book to identify whether the WNGF model shows similar patterns of advice bridging. The authors carried out the network analysis through questionnaires while also discussing the results and the underlying processes with managers to get a deeper understanding of the relations, hence we assume that they obtained correct information when assessing the advice network of the manufacturing company. Even if this cannot be thought of as ground truth it is still a valuable comparison. One of the key points is that there is high clustering in the network, due to most employees only receiving advice from others in their own country. We identify this clustering as well and show it through the coordinator role. Only two employees (ID: 70, 73) never become coordinator brokers. However, we can also see that the highest value for this role is at 0.41, suggesting that there is not one employee through whom all the advice flows, instead employees at the same location receive and advise each other. Therefore, the WNGF coordinator role highlights the importance of same-location advice brokering.

The other key observation is with regards to cross-country advice networks, where Cross & Parker state that the "only connections across countries were those of the leadership team along with a few relationships formed during past projects." We confirm these findings through the gatekeeper and representative roles. All global and local department managers, except for node 51, are among the top ten employees in terms of both roles. We further find that among the project leaders, there are also very influential representatives and gatekeepers, such as node 15, 54, and 18, while the few excellent researchers in terms of gatekeeping are 23 and 55, and in terms of representative are 28, 49, and 74. Thus again, we find similar results, where most cross-country advice flows through leadership level employees and a small number of researchers.

Secondly, we compare the extent to which the WNGF results correlate with the results obtained from dichotomization. We introduce a threshold of 35%, as this level removes all the smallest edges, with a weight of 1 (Very infrequent advice). This was the smallest possible threshold to be implemented due to the categorical nature of the data. Furthermore, the qualitative description of advice-giving also suggested that the removal of those links where advice is "very infrequent" may not distort the actual flow within the network. Secondly, we calculate the backbone network at $\alpha = 0.4$, the same level as in the previous application. Table 4 summarizes the Pearson and Spearman correlation between the five roles of WNGF and the five GF roles of the backbone and threshold networks. For all roles we find high correlation values, with the least strong values for the representative role. These strong correlations suggest that the WNGF measure does not deviate in its results from the results obtained from the dichotomized network, supporting the idea that the WNGF measure is also appropriate to identify the various brokerage roles. However, S2 Fig. further emphasizes through the shape of the ECDF

**Table 4. Correlation analysis of WNGF results with the backbone and threshold network on the organizational network.**

|  |  | Coordinator | Liaison | Itinerant | Gatekeeper | Representative |
|---|---|---|---|---|---|---|
| **Backbone** | **Pearson** | 0.931*** | 0.891*** | 0.914*** | 0.853*** | 0.597*** |
|  | **Spearman** | 0.911 *** | 0.684*** | 0.726*** | 0.771*** | 0.761*** |
| **Threshold** | **Pearson** | 0.961*** | 0.918*** | 0.976*** | 0.934*** | 0.635*** |
|  | **Spearman** | 0.956*** | 0.807*** | 0.888*** | 0.909*** | 0.796*** |

*** p< = .001

distributions that the WNGF roles provide the distributions most tending towards uniformity, in all roles except for the coordinator.

However, the largest differences arise for less embedded nodes that have smaller degree centrality, such as node 12, 14, 17, and 67. These are the nodes that appear to have the largest difference between the WNGF and GF values in various roles. Observing them on Fig 2 also highlights their small node size, which is proportional to their degree. While they are not the most important bridges in terms of advice networks, their role may still be important to note within the boundaries of their close co-workers. This is also reflected in the weight of these edges, especially for the three researcher nodes (14, 17, 67); many of their connections advise that they give advice at least somewhat frequently.

Although both applications show high correlation between the threshold method and the WNGF measure, the WNGF does not require any modifications to the networks or input values of parameters. On the other hand, the threshold method requires very different threshold levels—10% versus 35% for these two different macro and micro level datasets—to achieve reasonable results, which is arbitrary and sensitive to applications. As a result, with the same exact network data, two empirical analyses may produce very different results simply because different threshold levels are used. Therefore, our WNGF measure provides a benchmark result for studying the brokerage roles with weighted and directed networks, which is especially valuable for making comparisons across studies.

## Discussion and conclusion

In this paper we introduce the novel Weighted–Normalized Gould–Fernandez measure (WNGF) that can take edge weights into account when calculating the different GF brokerage roles. This is important in the context of the increasing availability of data for weighted networks and considering that many of the SNA brokerage metrics are not yet able to utilize the edge weight information. An exception for this is betweenness centrality, which is a global measure for brokerage that provides valuable insights by taking into account the whole network structure. For local measures such as GF brokerage, such extension is not yet available. The WNGF measure considers the varying group sizes by normalizing, and retains the full network structure by accounting for weighted edges. In the WNGF the focal node is identified as a broker if the sum of the inverse of the transaction flow between the focal node and either neighbor is smaller than the inverse of the flow directly from the incoming to the outgoing neighbor. In the context of a complete, weighted graph WNGF brokerage is not linked to bridging structural holes as by definition there are none in a complete graph. Instead, for such graphs, the measure provides a weaker definition of brokerage. While these brokers do not bridge holes in the network, they play important roles in connecting their neighbors. The WNGF provides a measure that can be applied to various types of networks. In the context of an incomplete, weighted graph, such as the organizational network analyzed in this paper, the measure will yield results based on the original definition of GF. This is because the absence of an edge is considered a 0, and taken into account as such when the edge weights are compared.

To test the validity and value-added of the WNGF, we apply both the WNGF and the dichotomization measures to two different empirical networks and carry out a correlation analysis. We first apply these measures to a trade network based on the EUREGIO data and then to an organizational network based on data collected by Cross and Parker [42]. Finally, the individual cases that present the most prominent divergence between the two measures are further analyzed. This final step has identified specific regions and employees that make the top five in the WNGF but that would not feature as prominent brokers when relying on dichotomization techniques.

The analysis of the EUREGIO dataset shows the importance of accounting for weights in a dense network. With the help of weights, brokerage position can be identified based on the strength of the tie. The correlation analysis between the dichotomized network and the WNGF results reveals a strong correlation. However, when we further investigate the differences, we see that while the backbone keeps the significant, regional edges, it still removes most instances of brokerage. On the less dense organizational network dichotomization without edge removal is also a possibility. While the correlation between WNGF and dichotomized results is still high, the largest differences reveal that the brokering possibilities of some employees who work within a small circle are overlooked even if they provide frequent advice. These results highlight that the WNGF generates valid results that are similar to those generated by dichotomized measures. They also show that through not altering the network one can gain further insights on brokerage. Lastly, we show that since no alterations and decisions must be made on how to dichotomize the network, the WNGF frees the user from the burden of making assumptions.

The benefits described above enrich the potential applications of the WNGF measure. In this paper we rely on an economic trade and organizational dataset for illustrative purposes, and for this reason we refrain from analyzing what the results of a WNGF brokerage analysis may mean for the regions or the employees in the company. There is large potential to further analyze the EUREGIO data in the context of regional studies. For example, in this context further analysis of brokerage roles using the WNGF can aid understanding of whether broker roles can help explain differences in the development of regions. The role of special regions, such as capital or port regions, can also be better understood. Other areas where this measure could be particularly useful is in organizational studies, where the role and performance of each employee could be better assessed if all the relevant information is considered.

Possible applications of WNGF span beyond regional and organizational studies and into all those contexts where retaining weights is important. Potential applications within weighted social datasets include email and collaboration networks or data where the frequency of the interaction is of relevance. With such data the information and innovation brokers, as well as potential bottlenecks, can be more rigorously identified. Such research may bring into light the difference between persisting or repeating edges compared to one-time interactions. WNGF can also be used to further analyze weighted information networks that measure how often people meet, talk, text, like, or retweet. In conclusion, we believe that the WNGF measure has a wide-ranging application across disciplines and different type of networks.

## Supporting information

**S1 Fig. Multi-measure comparison of the trade network through ECDFs of the WNGF and GF roles.**
(TIF)

**S2 Fig. Multi-measure comparison of the organizational network through ECDFs of the WNGF and GF roles.**
(TIF)

**S1 Table. EUREGIO data coverage on countries.**
(DOCX)

## Acknowledgments

We are grateful to Stefano Ghinoi, Francesca Pallotti, and Guido Conaldi for detailed comments and suggestions on an earlier version of the manuscript. The paper benefitted from comments at the University of Greenwich CBNA seminar.

## Author Contributions

**Conceptualization:** Zsófia Zádor, Zhen Zhu, Matthew Smith, Sara Gorgoni.

**Data curation:** Zsófia Zádor, Zhen Zhu.

**Formal analysis:** Zsófia Zádor.

**Methodology:** Zsófia Zádor, Zhen Zhu.

**Supervision:** Sara Gorgoni.

**Visualization:** Zsófia Zádor.

**Writing – original draft:** Zsófia Zádor, Zhen Zhu, Matthew Smith, Sara Gorgoni.

**Writing – review & editing:** Zsófia Zádor, Zhen Zhu, Matthew Smith, Sara Gorgoni.

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
