## [Decision Letter · Decision Letter 0]

6 Jul 2022

PONE-D-22-06397

A Weighted and Normalized Gould-Fernandez Brokerage Measure

PLOS ONE

Dear Dr. Zador,

Thank you for submitting your manuscript to PLOS ONE. After careful consideration, I feel that it has merit but does not fully meet PLOS ONE’s publication criteria as it currently stands. Therefore, I invite you to submit a revised version of the manuscript that addresses the points raised during the review process.

We look forward to receiving your revised manuscript.

Kind regards,

Luca De Benedictis, PhD

Academic Editor

PLOS ONE

Reviewers' comments:

Reviewer's Responses to Questions

**Comments to the Author**

1. Is the manuscript technically sound, and do the data support the conclusions?

Reviewer #1: Yes

Reviewer #2: Partly

2. Has the statistical analysis been performed appropriately and rigorously? 

Reviewer #1: Yes

Reviewer #2: Yes

3. Have the authors made all data underlying the findings in their manuscript fully available?

Reviewer #1: No

Reviewer #2: Yes

4. Is the manuscript presented in an intelligible fashion and written in standard English?

Reviewer #1: Yes

Reviewer #2: No

5. Review Comments to the Author

Reviewer #1: The paper extend the Gould and Fernandez (GF) local brokerage measure to account for weighted edges (WNGF). I think the topic is interesting. The authors shows that: (i) WNGF generates valid results, similar to those generated by dichotomised measures, (ii) only through not-altering the network can one gain a perspective on the details of brokerage, (iii) since no alterations and decisions have to be made on how to dichotomise the network, the WNGF frees the user from the burden of making assumptions. I think that the paper deserve publication, even if I suggest to perform some minor revision in order to improve it.

Comments:

• Page 5: I think that some references to the percolation literature (and, in particular, to percolation centrality measures) could be a due enrichment of the literature review. Moreover, another paper that could be cited is this: https://doi.org/10.1093/icc/dtab078

• Page 13: the authors normalize the metrics, but they explain too briefly why it is relevant. For instance, I can think that a non-normalized metric could be better because a broker (for instance a gatekeeper) in a large group could be more relevant than a broker in a small group. Please improve the comments in this part.

• Page 17: the fact that WNGF output is almost equal to the “threshold” method output seems to reduce the relevance of the WNGF. However, on the other hand, the fact that they are so similar can strengthen the reliability of WNGF and “threshold” and can imply that “threshold” method outperforms “backbone” one.

• Page 22 Table 5 (but also Table 3): can you try to explain why correlations of WNGF results with the backbone and threshold network is lower for “representative” nodes compared to the other brokerage roles?

Minor revisions:

• Page 6 “For example, in the context of the US Health policy domain’s communication network (11) by absorbing knowledge from another group and passing it to the other members of their own group;”: move this sentence to a footnote in order to avoid the break of the list.

• Check the citation style: it sometimes changes (for instance in page 7 “e.g.: Blau 1986…”). Moreover in page 7 there is a “Fernandez et al. 2000, 2000”.

• Page 21 Figure 1: node label color is almost indistinguishable. Please modify the colors.

• Section 4 is a bit lengthy with some repetitions. It could be a bit shortened.

• There are many typos. For instance:

o Page 2 line 34: “graphs The need” (add “.”)

o Page 7 line 136: “where different spatial levels;” do not use “;”, but “,”.

o Page 16 line 325: “self-loops.Hence” (add a space).

o Page 18 line 373: “without too much noise;” do not use “;”, but “,”.

o Page 18 line 374: “nuanced information;” do not use “;”, but “,”.

o Page 20 lines 407-422: it is all a repetition of the previous page.

o Page 25 line 515: “researchers could be observed;” do not use “;”, but “:”.

Reviewer #2: The authors report a proposal to extend brokerage network measures in presence of weighted directed network data. Two procedures for normalising and dichotomising network data have been also adopted. Real data examples are considered to show the usefulness of the proposal. The manuscript is of interest for network analysts but some major revisions are required to improve the readibility and to report the statistical analysis in a more appropriate and rigorous way.

Comments

Abstract

Improve the structure by specifying the added value of the proposed approach to deal with weighted directed networks and the main findings.

Introduction

- Improve the readibility of the contribution by better specify the main aim and justify the specific normalisation and dichotomisation procedures adopted among the ones proposed in the literature;

2. try to avoid repetitive sentences/words by rephrasing the theoretical framework (1.1) and the application fields (1.2) of brokerage measures;

Section 2

It could be better to introduce first the methodological approach proposed for the weighted version of the brokerage measures, and then to discuss the different procedures introduced for network data normalisation and dichotomisation. Among others, see also the threshold dichotomisation procedure discussed in Giordano and Primerano (2018), and the backbone procedure adopted in Genova et al. (2021) and the references therein.

Giordano, G., Primerano, I. The use of network analysis to handle semantic differential data. Qual Quant 52, 1173–1192 (2018)

Genova, V.G., Tumminello, M., Aiello, F. et al. A network analysis of student mobility patterns from high school to master’s. Stat Methods Appl 30, 1445–1464 (2021).

- Section 2.2 Check Equations 1, 2 and Table 1 in order to be more stable and rigorous adopting the same superscript for the broker node (i or j). In addition, more details are required to clarify the normalisation version of the GF brokerage roles described in the last column of Table 1 and in the Equation 2.

Section 3 Illustrative examples (Data)

Better clarify the added value in using WNGF measures given that the authors declare "We show that the WNGF measure produces similar results to that of the dichotomized networks but with more in-depth understanding of the role all nodes play." (page 19, 287-288)

3.1 Better describe Matrix Z for EUREGIO dataset and Table 2.

3.1.1. Report more details along the text about the results inserted in Appendix A (page 17, lines 342)

3.2 Improve the description of network information and the network visualisation of Figure 2.

Repetitive sentences are reported at page 20, from line 407 to 422 (see the same sentences at page 19, lines 390-406).

Conclusions

It is not clear the comments regarding computational social sciences and digital data for the scope of the present contribution.

References

Check all references along the text in line with the journal's style. Different styles are used to cite references (numbers or authors' last name and papers' publication year).

Minor remarks

Please clarify the following sentences:

- With the increase and abundance of data availability all these extra layers can be explored with the proper tool (page 8, 147-148)

- ...which then has to be dichotomised in order to analyse the different brokerage topologies, again pointing to the need for extending current measures to handle the large-scale data availability (pages 8-9, 168-169)

- we will first introduce the current dichotomisation practices through which the challenge of edge weights can be mitigated (page 9, 175-176);

Some sentences are repetitive along the text (e.g., nuanced information, nuanced features, nuanced results, nuanced analysis, etc.) and the contribution could be shorten.

It is suggested to rephrase the titles of the sections according to the main topics (Introduction, Theoretical background, Methodology, Data description and results, Discussion and Conclusions).

It is suggested to refine the use of English.

6. PLOS authors have the option to publish the peer review history of their article (what does this mean?). If published, this will include your full peer review and any attached files.

Reviewer #1: No

Reviewer #2: No

---

## [Author Response · Author response to Decision Letter 0]

26 Aug 2022

The Response to Reviewers has been attached to the submission as a separate file with detailed comments to each point raised.

---

## [Editor Report · Decision Letter 1]

30 Aug 2022

A Weighted and Normalized Gould‒Fernandez brokerage measure

PONE-D-22-06397R1

Dear Dr. Zador,

We’re pleased to inform you that your manuscript has been judged scientifically suitable for publication and will be formally accepted for publication once it meets all outstanding technical requirements. Please, improve the resolution of the Figures and the Tables so to make publication proof.

Kind regards,

Luca De Benedictis, PhD

Academic Editor

PLOS ONE

---

## [Editor Report · Acceptance letter]

6 Sep 2022

PONE-D-22-06397R1 

A Weighted and Normalized Gould‒Fernandez brokerage measure 

Dear Dr. Zádor:

I'm pleased to inform you that your manuscript has been deemed suitable for publication in PLOS ONE. Congratulations! Your manuscript is now with our production department. 

Kind regards, 

on behalf of

Dr. Luca De Benedictis 

Academic Editor

PLOS ONE